# Cutaneous Allodynia of the Withers in Cattle: An Experimental In Vivo Neuroanatomical Preliminary Investigation of the Dichotomizing Sensory Neurons Projecting into the Reticulum and Skin of the Withers—A Case Study on Two Calves

**DOI:** 10.3390/ani15121689

**Published:** 2025-06-06

**Authors:** Roberto Chiocchetti, Luciano Pisoni, Monika Joechler, Adele Cancellieri, Fiorella Giancola, Giorgia Galiazzo, Giulia Salamanca, Rodrigo Zamith Cunha, Arcangelo Gentile

**Affiliations:** Department of Veterinary Medical Sciences, University of Bologna, Via Tolara di Sopra, 50, 40064 Ozzano dell’Emilia, Italy; luciano.pisoni@unibo.it (L.P.); monika.joechler@unibo.it (M.J.); adele.cancellieri.vet@outlook.it (A.C.); fiorella.giancola0@gmail.com (F.G.); giorgiagali@gmail.com (G.G.); giulia.salamanca2@unibo.it (G.S.); rodrigozamithcunha@gmail.com (R.Z.C.); arcangelo.gentile@unibo.it (A.G.)

**Keywords:** bovine, Diamidino Yellow, dorsal root ganglion, Fast Blue, retrograde fluorescent tracer

## Abstract

Traumatic reticuloperitonitis, also known as “hardware disease”, is a common condition in cattle that can cause pain in the abdominal cavity. A clinical test called the Kalchschmidt pain test is used to detect this condition, based on the observation of skin hypersensitivity (cutaneous allodynia) at the withers. It is hypothesized that this reaction is caused by the convergence of visceral (from the reticulum) and somatic (from the skin) sensory information in the nervous system. This study investigated whether some sensory neurons located in the thoracic dorsal root ganglia (DRG) simultaneously send projections to both the reticulum and the skin of the withers, a condition known as dichotomization. Two retrograde fluorescent tracers were injected in two calves: Fast Blue (FB) into the reticulum and Diamidino Yellow (DY) into the skin of the withers. After 30 days, the DRG from T1 to T8 were examined under a fluorescence microscope. While many neurons were labeled from the skin, only a few were labeled from the reticulum, and no double-labeled neurons were found. These findings suggest that the convergence of visceral and somatic inputs likely occurs at a central level, rather than within individual sensory neurons, possibly involving vagal pathways and supra-spinal integration centers.

## 1. Introduction

First reported in 1948 [1] and then better detailed in 1954 by Kalchschmidt [2], cutaneous allodynia at the level of the withers (Head zone) has been used as common diagnostic evidence to confirm suspected hardware disease (hereinafter called “Kalchschmidt pain test”).

In fact, Kalchschmidt noted that animals affected by traumatic reticuloperitonitis experienced cutaneous hyperalgesia (allodynia) at the level of the withers. According to the definition proposed by Head in 1893 [3], at the base of the allodynia there is a supposed convergence of viscero- and somato-afferent (nociceptive) neurons at the same level of the spinal cord.

The “Kalchschmidt pain test” is carried out by gently lifting a skin fold at the level of the eighth thoracic spinous process, at the end of the inspiratory or at the beginning of the expiratory phase. In order to prevent peritoneal rub-induced pain (“withers pinch test”), attention should be paid to not inducing the animal to dip its spine. The presence of hyperalgesia is manifested by a few seconds of interruption of the respiratory phase (apnea); a first slight noise is produced by the closure of the glottis and a second one, in the form of a grunt, is produced by the abrupt re-opening of the glottis and the resumption of exhalation [1,2].

It is known that pain of the gastrointestinal apparatus is subtle and difficult to locate, and that this is often projected to body areas far from the algic site. Scientific evidence shows that nociceptive visceral afferents run together with efferent sympathetic fibers, and that the afferents which run in the parasympathetic nerves rarely carry pain information [4,5,6,7,8]. These findings indicated that the representation of a viscera within the central nervous system (CNS) might be quite inaccurate. The visceral nociceptive fibers, which originate from primary sensory neurons localized in the dorsal root ganglia (DRG), make synaptic contact with the secondary sensory neurons distributed in the dorsal horns of the spinal cord, a portion of which may also be simultaneously related to the somatic sensory fibers [9,10]. The spinal cord secondary sensory neurons send their axons to supra-spinal centers which integrate the visceral information in order to generate motor and sensory responses. The result is that pain perception may be attributed not to the affected viscera but to some other somatic areas distant from those viscera. Actually, an “interference” is created between the visceral algia and the somatic algia, which manifests itself with referred pain sensation. This physiological phenomenon is considered to be the basis of the clinical test proposed by Kalchschmidt for confirming suspected hardware disease. In this case, the visceral algia originates at the level of the reticulum (and its peritoneum), whereas the corresponding somatic area is at the level of the skin of the withers. A subliminal stimulus in the cutaneous area of the withers elicits the excitation of the neurons constituting the algogenic pathways shared by the visceral and the somatic afferents [1,2].

Currently, it is not clear whether the convergence of the visceral and the cutaneous sensitive information is processed before or after having entered the spinal cord. The first “hierarchical” level for the elaboration of the afferents might be represented by the DRG, the second by the spinal cord secondary sensory neurons, and the others by the higher encephalic structures, such as the thalamus [7]. The information is conveyed from the thalamus to the different brain regions involved in the somatosensory perception of pain [11]. Investigations carried out on rats [12] and cats [13] have shown that some DRG sensory neurons present a dichotomy of their peripheral processes; in practice, two divergent projections are created from the same nerve fibers for the skin and viscera. In addition, in laboratory rodents, it has been demonstrated that relevant percentages (up to 21%) of DRG neurons also dichotomize to reach two different viscera [6,14,15] or two different somatic areas [16,17,18]. Thus, it cannot be excluded that, in cattle, as verified in other smaller species, the first integration takes place at the level of DRG neurons. However, it is necessary to consider that, in the only double-labeling study found in the literature carried out on the relationship between the urinary bladder and the back skin, very few dichotomizing sensory neurons (0.05–0.1%) were identified in the DRG, even though these primary afferent neurons were housed in the same DRG [12]. Considering the available literature, one could hypothesize that even in cattle there could be a subclass of sensory neurons capable of simultaneously reaching two distinct anatomical areas with their divergent projections. Therefore, the aim of the present study was to identify the presence of DRG dichotomizing sensory neurons in cattle capable of innervating visceral (reticulum) and somatic skin areas (withers). This might indicate the pre-spinal convergence of visceral and cutaneous sensory information which is suspected to be the anatomical basis of the cutaneous allodynia of the withers present in the clinical situation of traumatic reticuloperitonitis (“Kalchschmidt pain test”).

## 2. Material and Methods

### 2.1. Study Design—Neuroanatomical Experimental Study with a Descriptive Observational Design

Inclusion Criteria: -Age between 30 and 120 days old.-Body weight ranging from 40 to 90 kg.-Both sexes (male and female).-Weaned and unweaned animals.-Clinically healthy at the time of enrollment.

Exclusion Criteria:-Presence of any comorbidities with clinical signs at the time of evaluation.-History of disease within 30 days prior to the study.-Participation in other clinical or experimental studies concurrently.-Administration of any medication within 15 days prior to enrollment.

Animals—Two male Holstein calves, one weaned (calf #1) and one unweaned (calf #2), weighing 78 and 44 kg, and of 90 and 40 days of age, respectively, were used in the study. The much younger age of the second calf was due to the technical difficulties experienced during surgery on the first calf, precisely in the approach to the reticulum during dorsal recumbency. In fact, in calf #1, the reticulum was difficult to access and the injections were difficult to handle due to the large development of the rumen and the depth of the viscera in the cranial abdomen (*regio hypocondriaca* and *regio xiphoidea*). Preoperative fasting was limited to 6 h in order to prevent hypoglycemia; however, access to water was maintained. The University of Bologna Animal Experimentation Ethics Committee approved all procedures (PROT.: 31331-X/10).

Anesthesia—Tranquilization/induction were performed by administering atropine sulfate (0.04 mg/kg, IM) (Fatro SpA, Ozzano dell’Emilia, Italy, Europe) and a mixture of xylazine hydrochloride (0.22 mg/kg, IM) (Nerfasin, Produlab Pharma BV, SJ Raamsdonksveer, the Netherlands, Europe) and ketamine (6 mg/kg, IM) (Nimatek, Eurovet Animal Health B V, Bladel, Noord-Brabant, the Netherlands, Europe). After recumbency and venous catheterization, deepening of the anesthetic plan was achieved by a rapid injection of thiopental sodium (8 mg/kg, IV) (Pentothal Sodium IV F 0.5 G 20 M, Intervet Productions Srl, Aprilia (LT), Italy, Europe).

Endotracheal intubation was performed while the calves were in sternal recumbency by means of laryngoscopy. An orogastric tube was also passed to promptly decompress the rumen and to allow the emission of the continuous gas formed, thereby avoiding ruminal meteorism during surgery. Anesthesia was maintained by intermittent positive pressure ventilation (IPPV) (Kontron Instruments ABT-4100 Anesthesia Respirator, San Diego, CA, USA) with E.T. 0.8–1.1% isoflurane (IsoFlo, ZOETIS SRL, Roma, Italy, Europe) in oxygen. Supportive fluid IV administration was carried out using dextrose–electrolyte solution to avoid hypoglycemia. Butorphanol tartrate (0.02 mg/kg IV) (Nargesicl, Richter Pharma AG, Wels, Austria, Europe) was administered previously as preemptive analgesia and during the surgical procedure by constant rate infusion (CRI) (24 µg/kg/h) [19].

The protocol included the preemptive administration of xylazine (α2-agonist which provides both sedation and visceral analgesia), ketamine (NMDA-antagonist with analgesic properties), and butorphanol tartrate (opioid k-agonist µ-antagonist, moderate analgesic). During the surgical procedure, which was limited to manipulations of intraabdominal viscera without any traumatic maneuvers, analgesia was managed by IV administration of butorphanol tartrate in CRI (0.02–0.05 mg/kg/H). The cutoff points for supplemental butorphanol, considering that the opioid lasts 1–2 h, were an increase in heart rate (HR) by >20–25% from baseline, increase in blood pressure by >20% from baseline, and the clinical signs represented by movements or muscle tension. Monitoring of the pulmonary and cardiovascular functions was carried out at 5 min intervals to maximize the safety of the anesthesia and improve the likelihood of an uneventful recovery (Appendix B). During the recovery period, analgesia was achieved by administering ketoprofen (3 mg/kg, IM) (Vet-Ketofen 10%, Merial, Toulouse, France, Europe); vital signs were continually monitored until the return of coughing, swallowing, and righting reflexes.

Surgical procedure—Surgery to access the reticulum was performed by a preumbilical midline celiotomy in dorsal recumbency (Figure 1a). As the reticulum was located deep in the abdomen it was necessary to bring it up manually for celiotomic access (Figure 1b). After a tracer injection, the abdominal wall was reconstructed using simple interrupted sutures.

### 2.2. Injection of the Fluorescent Tracers

Two different retrograde fluorescent tracers, Fast Blue (FB) (Sigma-Aldrich Chemie, Steinheim, Germany, Europe) and Diamidino Yellow (DY) (Sigma-Aldrich Chemie, Steinheim, Germany, Europe), which have affinity for two different portions of the neuronal cell body (cytoplasm and nucleus, respectively), were simultaneously injected into the reticulum wall and into the skin of the withers (Appendix A). The DRG were examined after 30 days for neurons labeled by FB and DY.

Injection of fluorescent tracer FB into the reticulum—The cranial face (facies diaphragmatica) of the exposed reticulum was slowly infiltrated with multiple-site injections (10 injections) of FB (2% aqueous solution) using 10 μL Hamilton syringes (Figure 1a,b). The needle was placed in a tangential position with respect to the surface of the reticulum in order to especially infiltrate the subperitoneal space. The syringe was gently withdrawn, and any tracer leakage was promptly removed from the serosa using a sterilized cotton swab.

Injection of fluorescent tracer DY into the skin of the withers—After abdominal surgery, the calves were positioned in sternal recumbency. After a trichotomy and preparation of the surgical field, the skin interposed between the vertebral process (processus spinosus) from T1 to T8 was slowly infiltrated with multiple-site injections (12 injections of 10 µL each; 2 injections for each intervertebral space, a few millimeters on the right and on the left of the median plane) of DY (2% aqueous solution) using 10 μL Hamilton syringes (Figure 1c).

Experimental period—The appropriate experimental period necessary for the tracer to reach the neurons of the thoracic DRG at the transport distance (about 80 cm) was determined on the basis of previous personal research experiences [20,21,22,23]. Therefore, for this study, an experimental period of 30 days was established.

Tissue preparation—At the end of the experimental period chosen, the calves were deeply sedated with acepromazine maleate (80 mcg/kg IM) (Prequillan, FATRO SpA, Ozzano dell’Emilia, Italy, Europe) and anesthetized with thiopental sodium (15 mg/Kg, IV) (Pentothal Sodium IV F 0.5 G 20 M, Itervet Productions Srl, Aprilia (LT), Italy, Europe) and were finally euthanized by means of the IV administration of embutramide mebenzonium iodide and tetracaine hydrochloride (Tanax, Intervet International GmbH, Unterschleissheim, Germany, Europe). After euthanasia, the thoracic portion of the spinal cord was immediately exposed. The T1–T8 spinal cord, surrounded by the dural sack, was immediately exposed in its full length by means of a dorsal laminectomy. The procedure carefully avoided cutting the spinal roots to ensure that the various segments of the spinal cord could be precisely identified and the DRG collected. The spinal cord was freed from the dura and divided into segments, localized by means of the spinal roots and by counting them from the first thoracic spinal nerve, located just caudal to the first rib. The spinal cord segments with the DRG attached were fixed for 36 h in 4% paraformaldehyde in phosphate buffer (0.1 M, pH 7.2) at 4 °C, rinsed overnight in phosphate-buffered saline (PBS: 0.15 M NaCl in 0.01 M sodium phosphate buffer, pH 7.2), and stored at 4 °C in PBS containing 30% sucrose and sodium azide (0.1%). The following day, the tissues were dipped in Tissue Tek^®^ (Sakura Finetek, Alphen aan den Rijn, Holland, Europe) mounting medium and stored at 4 °C overnight and were then frozen in isopentane cooled by liquid nitrogen. Longitudinal serial sections (14–16 µm thick) of both the left and the right DRG from T1 to T8 were cut on a cryostat and mounted on polylysine-coated slides. The sections (not coverslipped) were stored at −80 °C and subsequently examined on a fluorescent Nikon Eclipse Ni microscope (Nikon Instruments Europe BV, Amsterdam, the Netherlands, Europe) equipped with a filter system providing excitation light having a wavelength of 360 nm which elicits the blue (FB) and yellow (DY) fluorescent labeling of the neuronal cytoplasm and nucleus, respectively.

Analysis of the sections—The images were recorded using a Nikon DS-Qi1Nc digital camera and NIS Elements software BR 4.20.01 (Nikon Instruments Europe BV, Amsterdam, the Netherlands, Europe). Slight adjustments to contrast and brightness were made using Corel Photo Paint (Corel Photo Paint and Corel Draw, Ottawa, ON, Canada), whereas the figure panels were prepared using Corel Draw (Corel Photo Paint and Corel Draw, Ottawa, ON, Canada).

## 3. Results

Reticulum and skin injection sites—Fast Blue deposits were no longer recognizable beneath the serosal surface of the reticulum. On the contrary, circumscribed areas of yellow deposits of DY were observed within the thickness of the epidermis and derma of the skin of the withers, covering T1–T8 vertebral processes (Appendix A).

FB-labeled neurons—Only a few faintly labeled FB-positive neurons were observed in the caudal thoracic DRG. In calf #1, only eight FB-positive neurons were localized in the right T6 and T7 DRG; in particular, three neurons were in T6 DRG and five neurons in T7 DRG. In calf #2, only two FB-positive neurons were counted in the right T7 DRG (Figure 2a).

DY-labeled neurons—A large number of bright DY-labeled neuronal nuclei were observed in both calves. In calf #2, the greatest number of DY-labeled neurons was counted in the right and left T7 DRG in which 716 and 51 DY-labeled neurons were counted, respectively. Diamidino Yellow was also widely observed in the nuclei of the glial cells surrounding the DY-labeled sensory neurons (Figure 2b–f). Notably, there were DRG which totally lacked DY-positive neurons in both calves.

No neurons were observed with double labeling, i.e., neurons simultaneously displaying FB and DY within the cytoplasm and nucleus, respectively.

## 4. Discussion

FB labeling—The minimal sensory innervation of the calf reticulum, arising from thoracic DRG, was not expected. In fact, analyzing the literature concerning the sensory innervation of the ruminant gastrointestinal tract (GIT), a greater neuronal labeling of cattle DRG would have been expected, since the duodenal sensory innervation seemed to be relevantly represented in the thoracic DRG of sheep and cattle [24]. Ohomori et al. (2012) [24] observed retrogradely labeled neurons of ruminants distributed between T4 and L2 DRG, with a notable number of labeled sensory neurons in thoracic T4–T8. On the contrary, in sheep, the sensory innervation of the gastroduodenal junction, arising from the thoracic DRG, is modest [25]. However, it should be noted that, in this study, the gastroduodenal junction, or duodenum, was not analyzed; instead, the most cranial forestomach, which is mainly supposed to be under vagal efferent and afferent control, was analyzed [20,26]. Other studies, carried out on other species, have shown that the visceral sensory innervation running along the great splanchnic nerve runs from T1 to T12 DRG in the rabbit [27] and T1–T13 DRG in the cat [28]. Moreover, studies carried out on the sensory innervation of portions of the GIT, anatomically related to the tract injected in the present study, i.e., the caudal esophagus and the lower esophageal sphincter of the cat [29,30], showed retrogradely labeled neurons distributed in all the thoracic DRG.

An explanation accounting for the reduced number of FB-labeled sensory neurons could be the site of injecting the FB into the reticulum. In the present study, particular attention was devoted to injecting the solution into the space between the peritoneal serosa and the external musculature. Until recently, it has been thought that the painful sensitivity of the viscera lies in the stimulation of the peritoneum. Actually, as observed in an animal model (horse) with intestinal aganglionosis in which only the extrinsic innervation of the gut was visible [31], the majority of the extrinsic sensory nerve fibers were distributed in the mucosa and submucosa, while the serosa rarely contained sensory nerve fibers. Furthermore, Brookes et al. [32] characterized a type of “silent” nociceptor, which could be activated by local inflammation and which is mainly distributed along the large submucosal blood vessels. Although FB is a soluble tracer, it was separated from the *tunica submucosa* by two thick layers of smooth muscle, and it is possible that it did not come in contact with the submucosal afferent/nociceptive nerve fibers.

In general, visceral pain is difficult to locate for different reasons. The visceral information reaches the CNS by means of the following neuronal pathways: (1) sensory fibers associated with the vagus nerve, the cell bodies of which reside within the proximal and distal ganglia; (2) sensory fibers associated with the splanchnic nerves, the cell bodies of which reside in the thoraco-lumbar DRG, and (3) sensory fibers associated with the pelvic nerve, the cell bodies of which reside in the sacral DRG. The thoraco-lumbar DRG neurons seem to be the most responsible for transferring visceral pain sensations to the CNS, while the vagus nerve elicits sensations, such as satiety, nausea, and vomiting [33]. However, evidence exists showing that the vagal and pelvic nerves may also elicit pain [7]. Thus, it is evident that the CNS must process information which is transmitted from multiple nerve pathways and which may eventually lead to a certain “imprecision” of the signal, a more diffuse perception of pain.

Presumably, the sensory information, coming from the reticulum, requires a privileged nerve pathway, i.e., that of the vagus nerve. There are studies showing that, at least in sheep, the vagus nerve plays a crucial role in controlling the reticular groove, so much so as to even have a viscerotopic representation in its dorsal nucleus [20]. It should also be kept in mind that, in the forestomach, the enteric nervous system (ENS) (also called the “little brain”) regulates a variety of local functions and reflexes which is possible due to presence of intrinsic sensory neurons and that, in cattle, the highest ganglionic density was observed at the reticular groove [34]. This role of the vagus nerve and the presence of a well-developed ENS represents two plausible explanations for the paucity of thoracic extrinsic sensory neurons in this species. Another reason, however not yet demonstrated in cattle, is the presence of another type of sensory neuron, i.e., the intestinofugal neuron which has its cell body within the gut and sends its axons outside the gut into the sympathetic prevertebral ganglia [35].

DY labeling—In this study, the DY tracer was injected into the epidermis and dermis of the skin interposed between the T1–T8 vertebral processes. It is known that the skin hair of cattle shows a delicate network of nerve fibers [36]; thus, DY positivity was expected in all the DRG collected. Nevertheless, the study showed a total lack of DY-labeled neurons in some DRG. This evidence may indicate that the peripheral distribution of skin sensory fibers is not uniform or, more likely, that only injections of the tracer, close or in the bundles of sensory fibers, allowed labeling the DRG neurons. This drawback might also be explained by the physical properties of the DY tracer which has lower solubility and diffusibility than FB.

The main goal of this study was the identification of DRG dichotomizing sensory neurons which could also indicate a pre-spinal convergence of visceral and cutaneous sensory information in cattle. The absence of double-labeled DRG neurons, and the complexity of the visceral and somatic reactions evoked by the Kalchschmidt pain test, makes the involvement of the higher integration centers which integrate the afferent information to coordinate respiratory movements of the diaphragm (phrenic nerve), intercostal muscles (intercostal nerves), and larynx (vagus nerve) plausible. Thus, the vagus nerve, which controls the forestomach, the larynx, and perhaps the diaphragm (at least in the ferret) [37], seems to be the most involved nerve in the Kalchschmidt pain reaction. It is plausible that the solitary tract nucleus, in which visceral but also somatic information converges [38], might represent an integrative supra-spinal center in cattle.

The finding of DY-positive nuclei within numerous (but not all) satellite glial cells (SGCs) enveloping DY-positive neurons indicates that the tracer migrated through poorly specified neuron–glia junctions. Since DY is not a trans-synaptic tracer (DRG neurons do not interact with other DRG neurons or SGCs via synapses), it remains to be elucidated which type of transport occurs between neurons and SGCs. The presence of gap junctions between DRG neurons and SGCs, although not demonstrated, might be the most plausible explanation. Retamal et al. [39] described several types of connexin and pannexon channels and hemi-channels distributed on DRG sensory neurons and SGCs, and the possibility that the hemi-channels may join to form gap junctions cannot be ruled out. It has also been reported that inflammation enhances the transfer of a dye from neuronal cell bodies to the surrounding glia in the trigeminal ganglia, probably via gap junctions [40]. Braz et al. [41] demonstrated intra-ganglionic cell-to-cell communication, via the transfer of large molecules, which occurs between injured primary afferent neurons and satellite cells. In addition, a dye permeability assay showed that, under physiological conditions, SGCs in the same neuron–glia unit are highly coupled [42]. The latter finding may support the presence of several DY-positive satellite cells encircling the DY-labeled non-injured neurons observed in the present study. It is reasonable to consider that SGCs may modulate the activation of nociceptive neurons not only through the release of adenosine triphosphate (ATP) and other neurotransmitters/neuromodulators [42,43,44] but also through the uni- or bi-directional molecular passage between SGCs and sensory neurons.

Finally, it is appropriate to mention some innovative studies which open new scenarios, challenging the dogmas regarding the transmission of sensory information. Kung et al. [45] demonstrated that DRG neurons, in addition to receiving supra-spinal and intra-spinal inputs (acting on the central DRG neuronal process), may potentially also receive intra-ganglion modulation. They showed that DRG neurons may locally release glutamate and other transmitters which potentially modulate the activities of neighboring sensory neurons in an extra-synaptic manner. Additional studies are necessary and might indicate that referred pain could also be due to neurons innervating two different targets, which are, however, contiguous within the same DRG.

Limitation—The major limitation of this study is the use of only two calves; since it seemed evident that reticular sensory information traveled with the vagus nerve and not with the peripheral nerve fibers of DRG neurons, no other calves were utilized to obtain predictable information. In this way, the authors complied with the rule of the three “Rs” (replace, reduce, refine). Another limitation is certainly not having collected and observed the vagal sensory ganglia, with the aim of observing the presence of the neurons expressing FB labeling in their cytoplasm.

## 5. Conclusions

The absence of dichotomizing sensory neurons innervating both the reticulum and the skin of the withers in this preliminary study suggests that the cutaneous allodynia observed in traumatic reticuloperitonitis is unlikely to originate from a peripheral convergence at the level of the dorsal root ganglia. Instead, it supports the hypothesis that the integration of visceral and somatic nociceptive inputs involved in the Kalchschmidt pain response occurs at a central level, possibly involving the vagus nerve and other brainstem nuclei, as recently observed by Zhang et al. [46] who considered the neuronal pathway of gastric pain in rodents.

Although limited by the small sample size and the exclusion of vagal sensory ganglia from the analysis, this study opens new directions for understanding visceral pain and its somatic projection in cattle. Future research should investigate central integrative centers, particularly the role of the solitary tract nucleus and vagal pathways, to better understand referred visceral pain mechanisms and their implications for veterinary diagnosis.

## Figures and Tables

**Figure 1 animals-15-01689-f001:**
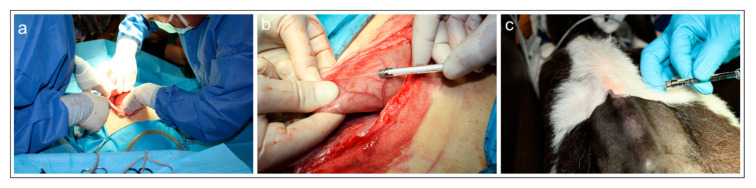
(**a**–**c**) Photographs of the surgery on the weaned calf (calf #1). (**a**) Surgical procedures during laparotomy. (**b**) Injection of fluorescent retrograde tracer Fast Blue into the reticulum. (**c**) Injection of fluorescent retrograde tracer Diamidino Yellow into the skin of the withers.

**Figure 2 animals-15-01689-f002:**
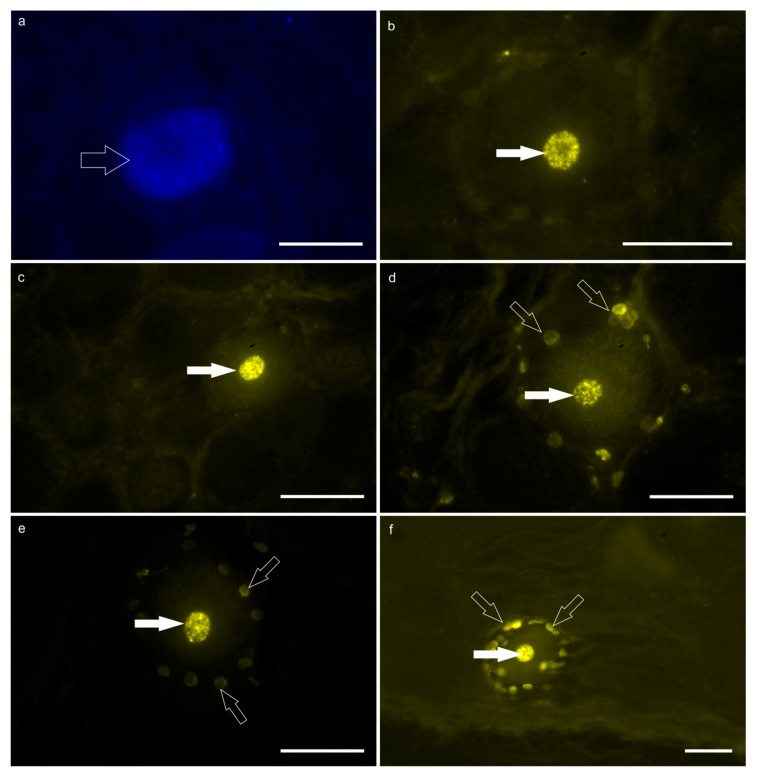
Fluorescent micrographs of the neurons of the T7 dorsal root ganglion (DRG) labeled with Fast Blue (FB; (**a**)) and Diamidino Yellow (DY; (**b**–**f**)), after injection of the tracers into the reticulum and the skin of the withers, respectively. (**a**) The open arrow indicates the FB-labeled cytoplasm of one DRG neuron. (**b**–**f**) The white arrows indicate the DY-labeled nuclei of five different DRG neurons. The open arrows (**d**–**f**) indicate the DY-labeled nuclei of the satellite glial cells surrounding the DY-positive sensory neurons. Scale bar: 50 µm.

## Data Availability

The raw data supporting the conclusions of this article will be made available by the authors on request.

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
