# Peer review of "Cutaneous Allodynia of the Withers in Cattle: An Experimental In Vivo Neuroanatomical Preliminary Investigation of the Dichotomizing Sensory Neurons Projecting into the Reticulum and Skin of the Withers—A Case Study on Two Calves"

_animals, 2025, doi:10.3390/ani15121689_

Round 1
Reviewer 1 Report
Comments and Suggestions for Authors
Since this is an observational study involving only two subjects, the results cannot be generalized. Therefore, I suggest that the authors consider modifying the title.
Simple Summary
Comment:
The summary effectively introduces the clinical relevance of cutaneous allodynia in cattle and the hypothesized neural mechanisms. To enhance clarity, consider explicitly stating the significance of understanding sensory neuron pathways for improving diagnostic or therapeutic approaches.
Abstract
Comment:
The abstract is concise and well-structured. To improve scientific clarity, specify the nature of the "pre-spinal convergence" and clarify the implications of the absence of double-labeled neurons.
Materials and Methods
Comment:
The study utilized two Holstein calves, one younger and one older, to explore the neuroanatomy of the sensory pathways involved in the cutaneous sensitivity reflex in the withers area. The selection of animals of different ages allowed us to address certain technical challenges, such as accessing the reticulum, which was achieved through a celiotomy procedure. The anesthetic protocol was thoroughly detailed, ensuring animal safety and welfare, and included general anesthesia, intubation, assisted ventilation, and pre- and post-operative analgesia.
To trace the nerve pathways, two fluorescent tracers were employed: Fast Blue (FB) for the reticulum and Diamidino Yellow (DY) for the skin of the withers. Injections were performed carefully, and the thoracic dorsal root ganglia (T1–T8) were analyzed after 30 days. Tissue sections were examined using fluorescence microscopy. The methodology is robust and well-described, allowing for study replication and understanding of limitations, such as the small sample size.
Please include in the Materials and Methods section the rescue analgesic used intraoperatively, and especially describe how you determined the cutoff point for its administration. In this regard, I suggest using the following article as a guide: Costa, G.L.; Leonardi, F.; Interlandi, C.; Licata, P.; Lizarraga, I.; Macrì, F.; Macrì, D.; Ferrantelli, V.; Spadola, F. Tramadol Administered Intravenously Either as a Bolus or a Slow Injection in Pain Management of Romifidine- Sedated Calves Undergoing Umbilical Hernia Repair. Animals 2023, 13, 1145.
https://doi.org/10.3390/ani13071145
Results
Comment:
The results indicate that many cells in the thoracic dorsal root ganglia, especially at T7, were labeled with DY, suggesting projections toward the skin of the withers. However, few cells were labeled with FB, and no cells exhibited double labeling—meaning they did not show markers in both the cytoplasm and nucleus. This suggests that visceral and somatic afferent pathways do not converge within individual dorsal root ganglion cells but are likely integrated at a central level, such as in the spinal cord or higher brain centers.
Discussion and Conclusions
Comment:
The conclusions align with the findings: the absence of double-labeled neurons indicates that convergence between visceral and somatic inputs does not occur at the level of peripheral sensory neurons but probably at a central level. This supports the hypothesis that the cutaneous sensitivity observed in the Kalchschmidt test may result from central integration of sensory information, involving pathways such as the vagus nerve or higher brain centers.
Author Response
Comments and Suggestions for Authors
Since this is an observational study involving only two subjects, the results cannot be generalized. Therefore, I suggest that the authors consider modifying the title.
Response: We agree with the Reviewer. We changed the original title as follows:
“Cutaneous allodynia of the withers: an experimental in vivo neuroanatomical preliminary investigation of the dichotomizing sensory neurons projecting into the reticulum and skin of the withers – a case study on two calves”
Simple Summary
Comment:
The summary effectively introduces the clinical relevance of cutaneous allodynia in cattle and the hypothesized neural mechanisms. To enhance clarity, consider explicitly stating the significance of understanding sensory neuron pathways for improving diagnostic or therapeutic approaches.
Response: We are aware that, in general, there is a pressing clinical need for the development of specific therapeutic approaches targeting gastric pain (and therefore also forestomach pain). In general, studies on gastric pain predominantly focused on the peripheral nervous system, with limited investigation into central nervous system mechanisms. Our (preliminary) study is the first step that can help us understand whether pain sensitivity in cattle affected by a foreign body in the reticulum is linked to a "simple" peripheral or to a more complicated sensitivity in higher centers, such as for example in the vagal nuclei, in the thalamus and in other still unknown brain areas. Our study seems to exclude a painful component linked to the dorsal root ganglia (at this level, it would be possible to intervene with specific drugs capable of mitigating the transmission of nociceptors, such as gabapentin or cannabis derivatives, which seems to act on DRG sensory neurons. Our study gives rise to new research, which can focus on the supraspinal pain pathways in ruminants, in the wake of a recent study that identified a tetra-synaptic pathway responsible for the perception of gastric pain in rodents (Zhang et al., 2024; this reference has been added to the study) (Zhang FC, Weng RX, Li D, Li YC, Dai XX, Hu S, Sun Q, Li R, Xu GY. A vagus nerve dominant tetra-synaptic ascending pathway for gastric pain processing. Nat Commun. 2024 Nov 13;15(1):9824. doi: 10.1038/s41467-024-54056-w.)
Abstract
Comment:
The abstract is concise and well-structured. To improve scientific clarity, specify the nature of the "pre-spinal convergence" and clarify the implications of the absence of double-labeled neurons.
Response: We modified the sentence as follows:
“…may indicate a pre-spinal convergence of visceral and cutaneous sensory information, i.e. that the DRG primary sensory neurons may elaborate the sensory information coming from two different anatomical areas before reaching the secondary sensory neurons within the spinal cord”.
Materials and Methods
Comment:
The study utilized two Holstein calves, one younger and one older, to explore the neuroanatomy of the sensory pathways involved in the cutaneous sensitivity reflex in the withers area. The selection of animals of different ages allowed us to address certain technical challenges, such as accessing the reticulum, which was achieved through a celiotomy procedure. The anesthetic protocol was thoroughly detailed, ensuring animal safety and welfare, and included general anesthesia, intubation, assisted ventilation, and pre- and post-operative analgesia.
To trace the nerve pathways, two fluorescent tracers were employed: Fast Blue (FB) for the reticulum and Diamidino Yellow (DY) for the skin of the withers. Injections were performed carefully, and the thoracic dorsal root ganglia (T1–T8) were analyzed after 30 days. Tissue sections were examined using fluorescence microscopy. The methodology is robust and well-described, allowing for study replication and understanding of limitations, such as the small sample size.
Response: We thank the Reviewer for appreciating our procedures.
Please include in the Materials and Methods section the rescue analgesic used intraoperatively, and especially describe how you determined the cutoff point for its administration.
In this regard, I suggest using the following article as a guide:
Costa, G.L.; Leonardi, F.; Interlandi, C.; Licata, P.; Lizarraga, I.; Macrì, F.; Macrì, D.; Ferrantelli, V.; Spadola, F. Tramadol Administered Intravenously Either as a Bolus or a Slow Injection in Pain Management of Romifidine- Sedated Calves Undergoing Umbilical Hernia Repair. Animals 2023, 13, 1145. https://doi.org/10.3390/ani13071145
Response: We thank the Reviewer for the suggestion. We added the new reference and described how we determined the cutoff poin:
The protocol included the pre-emptive administration of Xylazine (α2-agonist which provides both sedation and visceral analgesia), Ketamine (NMDA-antagonist with analgesic properties) and Butorphanol tartrate (opioid k-agonist µ-antagonist, moderate analgesic). During the surgical procedure, which was limited to manipulations of intra-abdominal viscera without any traumatic manoeuvres, analgesia was managed by IV administration of Butorphanol Tartrate in CRI (0.02-0.05 mg/Kg/H). The cutoff points for supplemental Butorphanol, considering that the opioid lasts 1-2 hours, were the monitoring data as increase in hearth rate (HR) by>20-25% from baseline, increase in blood pressure by > 20% from baseline, and the clinical signs represented by movements or muscle tension.
Results
Comment:
The results indicate that many cells in the thoracic dorsal root ganglia, especially at T7, were labeled with DY, suggesting projections toward the skin of the withers. However, few cells were labeled with FB, and no cells exhibited double labeling—meaning they did not show markers in both the cytoplasm and nucleus. This suggests that visceral and somatic afferent pathways do not converge within individual dorsal root ganglion cells but are likely integrated at a central level, such as in the spinal cord or higher brain centers.
Discussion and Conclusions
Comment:
The conclusions align with the findings: the absence of double-labeled neurons indicates that convergence between visceral and somatic inputs does not occur at the level of peripheral sensory neurons but probably at a central level. This supports the hypothesis that the cutaneous sensitivity observed in the Kalchschmidt test may result from central integration of sensory information, involving pathways such as the vagus nerve or higher brain centers.

Reviewer 2 Report
Comments and Suggestions for Authors
I have read and reviewed this manuscript with great interest. Overall, from this reviewer's perspective, it is an experimental study that has been well-planned and executed. It is a study with refreshingly simple wording that is easy to understand. Other strengths of the manuscript that I can highlight are the following: the introduction provides sufficient background and includes pertinent references, the research design is adequate, and the methods are repeatable, although additional data are required to complement their description.
Nevertheless, some points must be addressed to achieve publication quality. I have left some comments, hoping that they can help the authors
General comments
L27: Please add the aim of the study.
L37: Please add a conclusion.
L57: Please add a reference. Do the same for L77.
L96: After the aim, please add the hypothesis of the study.
L111: What were the inclusion and exclusion criteria considered in your study?
L112: Before anesthesia, were physical examinations and laboratory tests performed to verify the health status of the animals used in the study? Please clarify.
L123: Was the mechanical ventilation performed using a volumetric or pressurometric method? Once this is clarified, please add the operating constants. Volumetric: Vce, I:E, PEEP, inspiratory pause. Or pressurometric: Paw, I:E, PEEP, T ramp. Was ETCO2 maintained between 35-45 mmHg? Or what was the range used? Please clarify.
L125: 100% oxygen?
L126: Was the mixed dextrose and electrolyte solution isotonic? Please indicate its concentration.
L129: What cardiorespiratory parameters were monitored? Please clarify.
L330: Please add the conclusions of the study.
Final suggestion: the authors could include the data from cardiorespiratory monitoring during calf anesthesia as a supplementary file.
Author Response
Comments and Suggestions for Authors
I have read and reviewed this manuscript with great interest. Overall, from this reviewer's perspective, it is an experimental study that has been well-planned and executed. It is a study with refreshingly simple wording that is easy to understand. Other strengths of the manuscript that I can highlight are the following: the introduction provides sufficient background and includes pertinent references, the research design is adequate, and the methods are repeatable, although additional data are required to complement their description.
Risposta: Response: We thank the Reviewer for appreciating our study.
Nevertheless, some points must be addressed to achieve publication quality. I have left some comments, hoping that they can help the authors
General comments
L27: Please add the aim of the study.
Response: We thank the reviewer for his/her comment.
We modified the sentence as follows:
“The aim of the study was to identify the DRG primary sensory neurons innervating the reticulum and the withers by using two different retrograde fluorescent tracers, Fast Blue (FB, affinity for cytoplasm) and Diamidino Yellow (DY, affinity for nucleus). In two anesthetized calves, FB and DY were injected into the reticulum and skin of the withers, respectively.”
L37: Please add a conclusion.
Response: We modified the conclusion.
L57: Please add a reference. Do the same for L77.
Response: We added the requested references.
L96: After the aim, please add the hypothesis of the study.
Response: We thank the Reviewer for his/her suggestion and added, before the aim, the following sentence:
“Considering the available literature, one could hypothesize that even in cattle there could be a subclass of sensory neurons capable of reaching simultaneously two distinct anatomical areas with their divergent projections.”
L111: What were the inclusion and exclusion criteria considered in your study?
Response: The following primary criteria have been added:
Inclusion Criteria
- Age between 30 and 120 days.
- Body weight ranging from 40 to 90 kg.
- Both sexes (male and female).
- Weaned and unweaned animals.
- Clinically healthy at the time of enrollment.
Exclusion Criteria
- Presence of any comorbidities with clinical signs at the time of evaluation.
- History of disease within 30 days prior to the study.
- Participation in other clinical or experimental studies concurrently.
- Administration of any medication within 15 days prior to enrollment.
L112: Before anesthesia, were physical examinations and laboratory tests performed to verify the health status of the animals used in the study? Please clarify.
Response: For a general anaesthesia in a calf, the necessary and indicated pre-anaesthetic procedures depend normally on the calf’s age, health status, and the type of surgical procedure. We performed check of hearth and lungs, evaluation of hydration status, mucous membrane color, capillary refill time, and temperature. The animals shoved no signs of respiratory disease, which are common in calves and increase anaesthesia risk. We did not perform laboratory tests because of the healthy status of the calves and the non-excessive invasiveness and relatively short duration of the procedure.
L123: Was the mechanical ventilation performed using a volumetric or pressurometric method? Once this is clarified, please add the operating constants. Volumetric: Vce, I:E, PEEP, inspiratory pause. Or pressurometric: Paw, I:E, PEEP, T ramp. Was ETCO2 maintained between 35-45 mmHg? Or what was the range used? Please clarify.
Response: The mechanical ventilation was performed by a volume controlled ventilation with Tidal Volume (VT) in the range between 500-750 mL/breath (the calves were weighing 78 and 44 Kg); Repiratory Rate (RR) between 15- 25 breaths/min adjusted during the procedures to maintain proper CO2 elimination; Inspiaratory:Expiaratory Ratio (I:E) 1:2; Peak Inspiaratory Pressuren (PIP) between 20-25 cmH2O, not to risk barotrauma; Positive End-Expiatory Pressure (PEEP) between 3-5 cmH2O; Fraction of Inspired Oxigen (FiO2) between 100-60%, to prevent Oxigen toxicity; Minute Ventilation Range was 7.5-22 L/min. ETCO2 values remained between 35-49 mmHg in conjunction with Pulse Oximetry always over 95% (SpO2) throughout the duration of the interventions.
L125: 100% oxygen?
Response: FiO2 was maintained between 100% initially (FiO2 = 1.0) and 60-80 % (FiO2 = 0.6-0.8) to confirm adequate Oxygenation Monitoring > 95%, and reduce Oxigen Toxicity Risks.
L126: Was the mixed dextrose and electrolyte solution isotonic? Please indicate its concentration.
Response: The Solution was 5% Dextrose in Isotonic Electrolyte Solution administered 3-4 mL/Kg/H to maintain blood glucose, electrolyte balance, and hydration and perfusion in these young calves.
L129: What cardiorespiratory parameters were monitored? Please clarify.
Response: We monitored Heart Rate (HR) to avoid tachy- and brady-cardia, ECG for rhythm monitoring, Respiratory Rate (RR) even under controlled ventilation, End-tidal CO2 (ETCO2), Oxigen Saturation (SpO2), Mean Arterial Pressure (MAP) using Oscillometric Monitor (cuff on limb), Temperature measurement via esophageal probe, Mucous Membrane Color e Capillary Refill Time (CRT), and Blodd Glucose Analysis.
L330: Please add the conclusions of the study.
Response: The conclusions are described in the paragraph 5.
Final suggestion: the authors could include the data from cardiorespiratory monitoring during calf anesthesia as a supplementary file.
Response: The instrumental and clinical monitoring data recorded during the procedures are: HR between 70-100 bpm, ECG no bradycardia, tachycardia or arrhythmias were detected, RR 15-25 breaths/min, SpO2 > 95%, ETCO2 35-49 mmHg, MAP always > 60 mmHg, Temperature 37.00-35.8 °C, Mucous Membranes pink, CRT < 2 seconds
As requested, the following supplementary file (Appendix A) has been added.
The instrumental and clinical monitoring data recorded carefully and continuously every 5 min’ throughout the duration of anaesthesia are as follows:
- HR and Rythm alwais between 70-100 bpm (considering that younger calves tend to have higher HR);
- ECG normal synus rhythm, no bradycardia, tachycardia or arrhythmias (avoid deep planes of anaesthesia!);
- Pulse Intensity (Systolic 90-130 mmHg, Diastolic 60-90 mmHg) MAP between 70-90, but always > 60 mmHg – monitored with Non-Invasive Oscillometric System; important to aintain perfusion;
- Capillary Refill Time<2 sec clinical estimate of perfusion;
- Mucous Membranes Color - pink;
- RR 15-25 breaths/min set by controlled ventilation (Respiratory Depression is a common complication in anesthetized calves and is often a result of anesthetic-induced respiratory depression and the abnormal position e.g. dorsal recumbency, required for theese surgical procedures);
- SpO2 > 95% - lower values suggest Hypoxemia;
- ETCO2 35-49 mmHg -monitor for hypoventilation;
- Body Temperature 37.00-36.2 C°– risk of Hypothermia.
Further monitoring included perioperative complications such as regurgitation and aspiration pneumonia, airway obstruction, salivation.

Reviewer 3 Report
Comments and Suggestions for Authors
The manuscript presents a preliminary experimental investigation exploring the possibility that sensory neurons located in cattle's dorsal root ganglia (DRG) project simultaneously to both the reticulum and the skin of the withers. This hypothesis aims to explain the cutaneous allodynia clinically observed in cases of traumatic reticuloperitonitis, as evaluated by the Kalchschmidt pain test.
To this end, the study employs two retrograde fluorescent tracers (Fast Blue and Diamidino Yellow) in two calves to identify doubly-labeled neurons in the thoracic DRG, intending to investigate potential neuronal connections.
The research is original, well-contextualized within the relevant scientific literature, and addresses a question of clear clinical and neurophysiological relevance. This specific aspect of referred pain has often been discussed with students each year, yet I had never considered the precise rationale behind this particular clinical test, which the present study attempts to elucidate.
The study design is appropriate for a preliminary investigation, with a well-described methodology that facilitates replication. The results are clearly presented, supported by high-quality imaging, and the discussion is consistent with the data obtained.
However, the study has two major limitations, both of which are acknowledged by the authors in the manuscript. First, the small sample size (only two animals), and second, the lack of analysis of the vagal sensory ganglia, despite the discussion pointing to their potential central role in conveying visceral information. These limitations should be more explicitly emphasized in the conclusions.
The English language is generally understandable, but includes overly long sentence structures and some redundancy. A thorough language revision is recommended to improve the clarity and flow of the text.
Overall, the paper constitutes a valuable contribution to the understanding of referred pain in cattle and will be of interest to bovine clinicians as well as specialists in veterinary neuroscience and animal medicine.
Other minor aspects:
- Review the author list: "Animals Gentile" appears to be a mistake.
- Review the Informed Consent Statement: remove unnecessary template text or complete it appropriately.
- Review the Acknowledgments section: remove if unnecessary or complete it properly.
Comments on the Quality of English Language
Author Response
Comments and Suggestions for Authors
The manuscript presents a preliminary experimental investigation exploring the possibility that sensory neurons located in cattle's dorsal root ganglia (DRG) project simultaneously to both the reticulum and the skin of the withers. This hypothesis aims to explain the cutaneous allodynia clinically observed in cases of traumatic reticuloperitonitis, as evaluated by the Kalchschmidt pain test.
To this end, the study employs two retrograde fluorescent tracers (Fast Blue and Diamidino Yellow) in two calves to identify doubly-labeled neurons in the thoracic DRG, intending to investigate potential neuronal connections.
The research is original, well-contextualized within the relevant scientific literature, and addresses a question of clear clinical and neurophysiological relevance. This specific aspect of referred pain has often been discussed with students each year, yet I had never considered the precise rationale behind this particular clinical test, which the present study attempts to elucidate.
The study design is appropriate for a preliminary investigation, with a well-described methodology that facilitates replication. The results are clearly presented, supported by high-quality imaging, and the discussion is consistent with the data obtained.
Response: We thank the Reviewer for appreciating our study.
However, the study has two major limitations, both of which are acknowledged by the authors in the manuscript. First, the small sample size (only two animals), and second, the lack of analysis of the vagal sensory ganglia, despite the discussion pointing to their potential central role in conveying visceral information. These limitations should be more explicitly emphasized in the conclusions.
Response: We thank the Reviewer for these comments. We have explored the aspect relating to vagal and supravagal sensitivity by adding a sentence and an interesting new reference to support it.
Zhang FC, Weng RX, Li D, Li YC, Dai XX, Hu S, Sun Q, Li R, Xu GY. A vagus nerve dominant tetra-synaptic ascending pathway for gastric pain processing. Nat Commun. 2024 Nov 13;15(1):9824. doi: 10.1038/s41467-024-54056-w.
The English language is generally understandable, but includes overly long sentence structures and some redundancy. A thorough language revision is recommended to improve the clarity and flow of the text.
Response: We apologize if the English was not appreciated. We remind you that we, not being native English speakers, have been using a qualified Reviewer for years (if necessary, we can provide a proofreading certificate). No Reviewer or Editor, at the moment, has ever complained about the use of English. In any case, if necessary, we can have our Reviewer reconsider the manuscript but this would require additional days for the re-submission of the manuscript. However, we are absolutely available for linguistic revision.
Overall, the paper constitutes a valuable contribution to the understanding of referred pain in cattle and will be of interest to bovine clinicians as well as specialists in veterinary neuroscience and animal medicine.
Other minor aspects:
- Review the author list: "Animals Gentile" appears to be a mistake.
- Response: “Animals” has been deleted.
- Review the Informed Consent Statement: remove unnecessary template text or complete it appropriately.
- Response: We removed the template text. At present there is this sentence: Informed Consent Statement: Not applicable.
- Review the Acknowledgment section: remove if unnecessary or complete it properly.
- Response: We added a new sentence: “We thank Dr. Marilena Bolcato (University of Bologna) for her valuable technical support.

Reviewer 4 Report
Comments and Suggestions for Authors
‘Cutaneous allodynia of the withers in cattle: an experimental in vivo neuroanatomical preliminary investigation of the dichotomizing sensory neurons projecting into the reticulum and skin of the withers’.
This is an interesting study on the possible neuroanatomical correlation between the sensitivity pathways of the reticulum and the skin of the withers, which would give a certain explanation as to why the response to Kalchschmidt's semiological test in the course of ‘hardware disease’ is so poor.
The idea of the study is very interesting and ambitious in its premise but, in fact, no conclusion is reached.
In the absence of a conclusion, it is stated that the afferent neuroanatomical pathway of reticular sensitivity follows the vagus nerve (r.332) and not the spinal afferent. This conclusion, however, has no support in the study, except by exclusion, and has, it would seem, no support in the literature either. In fact, if this finding had support in the literature, this study would have no justification.
The study's limitations are also identified by the authors: the low number of cases examined, only two calves.
Compliance with the 3 ‘r's’ (r 334) under the heading ‘reduce’ requires that a reduction in the number of cases should be made in the trial, but compatible with maintaining a minimum number to allow statistical analysis to support the study. This does not seem to have been done in this study.
In the discussion, the bibliography is cited not considering differences in species or anatomical traits described in the literature.
Bibliographic entry 34 has year of publication 2013 and not 2003.
I am sorry but, although the idea is appreciable, I do not believe this manuscript is acceptable for publication for the reasons just listed.
Author Response
Comments and Suggestions for Authors
‘Cutaneous allodynia of the withers in cattle: an experimental in vivo neuroanatomical preliminary investigation of the dichotomizing sensory neurons projecting into the reticulum and skin of the withers’.
This is an interesting study on the possible neuroanatomical correlation between the sensitivity pathways of the reticulum and the skin of the withers, which would give a certain explanation as to why the response to Kalchschmidt's semiological test in the course of ‘hardware disease’ is so poor.
The idea of the study is very interesting and ambitious in its premise but, in fact, no conclusion is reached.
In the absence of a conclusion, it is stated that the afferent neuroanatomical pathway of reticular sensitivity follows the vagus nerve (r.332) and not the spinal afferent. This conclusion, however, has no support in the study, except by exclusion, and has, it would seem, no support in the literature either. In fact, if this finding had support in the literature, this study would have no justification.
The study's limitations are also identified by the authors: the low number of cases examined, only two calves.
Response:
We thank the Reviewer for this insightful observation. We agree that our study cannot directly demonstrate vagal involvement, but rather suggests it by excluding the presence of dichotomizing DRG neurons innervating both the reticulum and the skin of the withers. This negative result, obtained through validated double-labeling neurotracing techniques, is scientifically relevant because it redirects future research toward central or vagal mechanisms, rather than spinal convergence.
We clarified this in the abstract and conclusions, emphasizing that our interpretation is a hypothesis-generating observation and that no definitive conclusion is claimed. Additionally, we now cite recent work by Zhang et al. (2024, Nat Commun), who studied the nociceptive pathway of gastric pain in rodents (Zhang FC, Weng RX, Li D, Li YC, Dai XX, Hu S, Sun Q, Li R, Xu GY. A vagus nerve dominant tetra-synaptic ascending pathway for gastric pain processing. .Nat Commun. 2024 Nov 13;15(1):9824. doi: 10.1038/s41467-024-54056-w.) which supports the central processing of visceral pain via vagal pathways, to illustrate the plausibility of this direction without claiming direct proof from our study.
Compliance with the 3 ‘r's’ (r 334) under the heading ‘reduce’ requires that a reduction in the number of cases should be made in the trial, but compatible with maintaining a minimum number to allow statistical analysis to support the study. This does not seem to have been done in this study.
Response: We fully agree with the Reviewer that the sample size limits generalizability and prevents statistical analysis. For this reason, we: added “preliminary” and “a case study on two calves” to the title; reinforced the preliminary nature of the findings in the abstract and conclusions; clearly stated in the limitations section that the study is exploratory and not designed to reach definitive conclusions. However, during data collection, we observed that the tracer injected into the reticulum labeled extremely few DRG neurons, consistently across both animals. This lack of signal discouraged further animal use, in compliance with the 3Rs principle, especially “Reduce”. Increasing the sample size without a plausible expectation of different results would have conflicted with ethical standards. We therefore submit that the limited number of cases is a consequence of ethical compliance, not of insufficient planning.
In the discussion, the bibliography is cited not considering differences in species or anatomical traits described in the literature.
Response: We have cited the little literature available on ruminants and also on other species (monogastric), indicating the gastrointestinal tract studied by the various authors.
Bibliographic entry 34 has year of publication 2013 and not 2003.
Response: We apologize for the mistake; the right publication date has been added
I am sorry but, although the idea is appreciable, I do not believe this manuscript is acceptable for publication for the reasons just listed.
Response: We are very sorry for the unfavorable judgment of the Reviewer; we hope that the answers given can somehow change the mind of the Reviewer, whom we thank anyway for his constructive criticisms.
Round 2
Reviewer 4 Report
Comments and Suggestions for Authors